# Retarding oxidation of copper nanoparticles without electrical isolation and the size dependence of work function

G. Dinesha M.R. Dabera [1], Marc Walker [2], Ana M. Sanchez [2], H. Jessica Pereira [1], Richard Beanland[2] & Ross A. Hatton [1]

Copper nanoparticles (CuNPs) are attractive as a low-cost alternative to their gold and silver analogues for numerous applications, although their potential has hardly been explored due to their higher susceptibility to oxidation in air. Here we show the unexpected findings of an investigation into the correlation between the air-stability of CuNPs and the structure of the thiolate capping ligand; of the eight different ligands screened, those with the shortest alkyl chain, $-(CH_2)_2-$, and a hydrophilic carboxylic acid end group are found to be the most effective at retarding oxidation in air. We also show that CuNPs are not etched by thiol solutions as previously reported, and address the important fundamental question of how the work function of small supported metal particles scales with particle size. Together these findings set the stage for greater utility of CuNPs for emerging electronic applications.

[1] Department of Chemistry, University of Warwick, Gibbet Hill Road, Coventry CV4 7AL, UK. [2] Department of Physics, University of Warwick, Gibbet Hill Road, Coventry CV4 7AL, UK. Correspondence and requests for materials should be addressed to R.A.H. (email: Ross.Hatton@warwick.ac.uk)

Gold (Au) and silver (Ag) nanoparticles (NPs) are key components of hybrid materials for emerging electronic devices based on organic[1] and perovskite semiconductors[2] as well as plasmonic hot-electron devices[3] and nanoelectrodes for molecular electronics[4]. However, for utilisation in large area, price sensitive applications such as photovoltaics, the high cost of Au and Ag is prohibitive. Copper (Cu) NPs are an attractive alternative because Cu is ~1% of the metal cost[5] whilst offering a comparable electrical and thermal conductivity, in conjunction with a narrow, localised surface plasmon resonance (LSPR) absorption at a wavelength between that of Ag and Au, and of comparable intensity[6]. The drawback of Cu is its higher susceptibility to oxidation in air, and so its potential for the aforementioned applications has been sparsely investigated.

Both $Cu_2O$ and CuO are semiconductors with accessible band edges for charge carrier injection/extraction[7] combined with high hole mobilities[8,9], and so the formation of a thin oxide layer at the NP surface does not necessarily impede charge transfer between the metal core and its surroundings, although it does degrade the optical properties by damping the LSPR response[6]. Solution processable metal NPs are invariably synthesised by reduction of a metal salt in the presence of organic ligand molecules, and so the ligand is the primary determinant of the extent to which the metal core is electrically coupled to its surroundings. Saturated chain hydrocarbons are the ligand type of choice for CuNP synthesis because they facilitate NP growth with a tight size and shape distribution, and render the NPs processable from solution. However, since the mechanism of charge transport through a saturated organic molecule is off-resonance tunnelling[10], a ligand length of >2 nm is sufficient to effectively electrically insulate the NP from its surroundings[10,11]. Indeed monolayers of saturated molecules with lengths greater than ~2 nm have proved to be remarkably effective gate insulators in organic transistors when used in conjunction with a nano-thickness inorganic dielectric layer[10,11]. Conversely, saturated small molecules with lengths <1 nm have been widely used in organic electronics to tune the interfacial energetics and surface energy at the electrode/organic semiconductor interface in organic light-emitting diodes[12], organic photovoltaics[13–15] and transistors[16], without significantly adding to device series resistance. Solution processable organic semiconductors such as poly(3-hexylthiophene-2,5-diyl) and phenyl-$C_{61}$-butyric acid methyl ester also tolerate insulating molecular chains of 3–6 $CH_2$ units attached to the conjugated core. Experience therefore shows that, as a rule-of-thumb, short saturated organic chains can be accommodated in electronic devices without significantly adversely impacting device series resistance, provided the length is less than ~1 nm. In the context of CuNPs the ligand can, in principle, also help passivate the metal core towards oxidation, although literature reports relating to the correlation between CuNP stability in air and the structure of the capping ligand are sparse in number and limited to the use of ligands based on alkyl chains >1 nm in length[17–19]. Given that the gradual ingress of oxygen and water into electronic devices is inevitable even with encapsulation[20] there is a need for the identification of organic capping ligands that retard oxidation sufficiently for the intended application, without electrically insulating the metal core from its surrounding matrix. Surprisingly, of the very small number of reports pertaining to the stability of CuNPs in air, all but one[21] focus on investigating stability of CuNPs in solution, which is not the relevant phase for electronic applications[17–19].

In principle thiol moieties are the anchor group of choice for binding organic ligands to the surface of CuNPs because the Cu-thiolate bond is a strong covalent linkage[11] and can displace more weakly bound ligands such as amine and carboxylic acid, to achieve a ligand layer matched to the intended application.

However, it has been reported that thiol solutions can also etch CuNPs[18], which would render this approach to ligand exchange unusable. Furthermore, the way that the work function ($\varphi$) of supported metal NPs scales with size for sizes greater than that at which quantum size effects dominate; >2 nm, is also a contentious issue[22–28]. These gaps in the understanding of how to modify the surface of CuNPs to match the application, and how the properties of small metal NPs scale with size need to be addressed if CuNPs are to achieve their potential as a substitute for Au and Ag for emerging electronic applications.

Herein we show that primary amine ligands on CuNPs can be exchanged with thiols without etching the CuNP, and report the unexpected findings of an investigation into the stability of isolated CuNPs towards oxidation in air when capped with the eight different thiolate ligands. The experiment design is distinct from that used to study the colloidal CuNP stability in all other literature reports, since it is based on monitoring (in real time) the oxidation of largely isolated CuNPs tethered to a solid substrate via the evolution of the LSPR band—a direct approach which simplifies the interpretation of the data. Finally, ligand-capped CuNPs are used to elucidate the correlation between the $\varphi$ and size of small spherical metal particles, using Kelvin probe force microscopy (KPFM). Collectively these three important insights provide a framework for the utility of CuNPs for emerging electronic applications.

## Results

**Copper NP synthesis and characterisation.** CuNPs capped with oleylamine were synthesised using a method reported by Grouchko et al.[29]. Figure 1a, b (and Supplementary Fig. 1) shows representative Annular Dark Field Scanning Transmission Electron Microscope (ADF-STEM) images of oleylamine-capped CuNPs, from which it is evident that they are approximately spherical with a distribution of diameters in the range 4–22 nm, and are highly crystalline consisting of both mono and polycrystalline NPs. Individual Cu atoms are also visible at the edge of a NP in Fig. 1a that have been dislodged from the NP surface under the electron beam, which is known to occur at high magnification[30]. The inset in Fig. 1c corresponds to the Fast Fourier Transform (FFT) of the area enclosed in the yellow square in Fig. 1a. This FFT is in agreement with the 110 zone axis, containing the (111) and (002) of the bulk Cu with a face-centred cubic (fcc) structure (space group: $Fm$-$3m$) when a lattice constant of 361.5 pm is assumed[18,31,32].

The X-ray diffraction (XRD) pattern of a thick film of CuNPs shown in Fig. 1c confirms the high degree of crystallinity of the bulk (~95%) of CuNPs in the distribution: the reflections corresponding to Cu (111), Cu (200), Cu (220), Cu (311) and Cu (222) planes are assigned[33], although the possibility that the small population of particles (~5%) with diameters less than ~4 nm may have lower crystallinity cannot be ruled out on the basis of these measurements. The weak and broad reflection at ~37° is attributed to $Cu_2O = 36.5°$[20] since the samples were briefly exposed to ambient air when transferring from the nitrogen-filled glove box to the nitrogen-filled graphite dome that houses the sample inside the XRD chamber. XPS and Auger spectroscopy analysis confirm that freshly prepared oleylamine-capped and ligand-exchanged CuNPs are essentially free of oxides (Supplementary Figs. 2 and 3 and Supplementary Table 1).

**Ligand exchange.** Ligand exchange reactions on metal NPs are invariably performed in solution[17,18,34,35]. Here ligand exchange was performed on a sub-monolayer of CuNPs tethered to the surface of glass and silicon substrates modified with a monolayer of (3-mercaptopropyl)trimethoxysilane (MPTMS) as depicted in

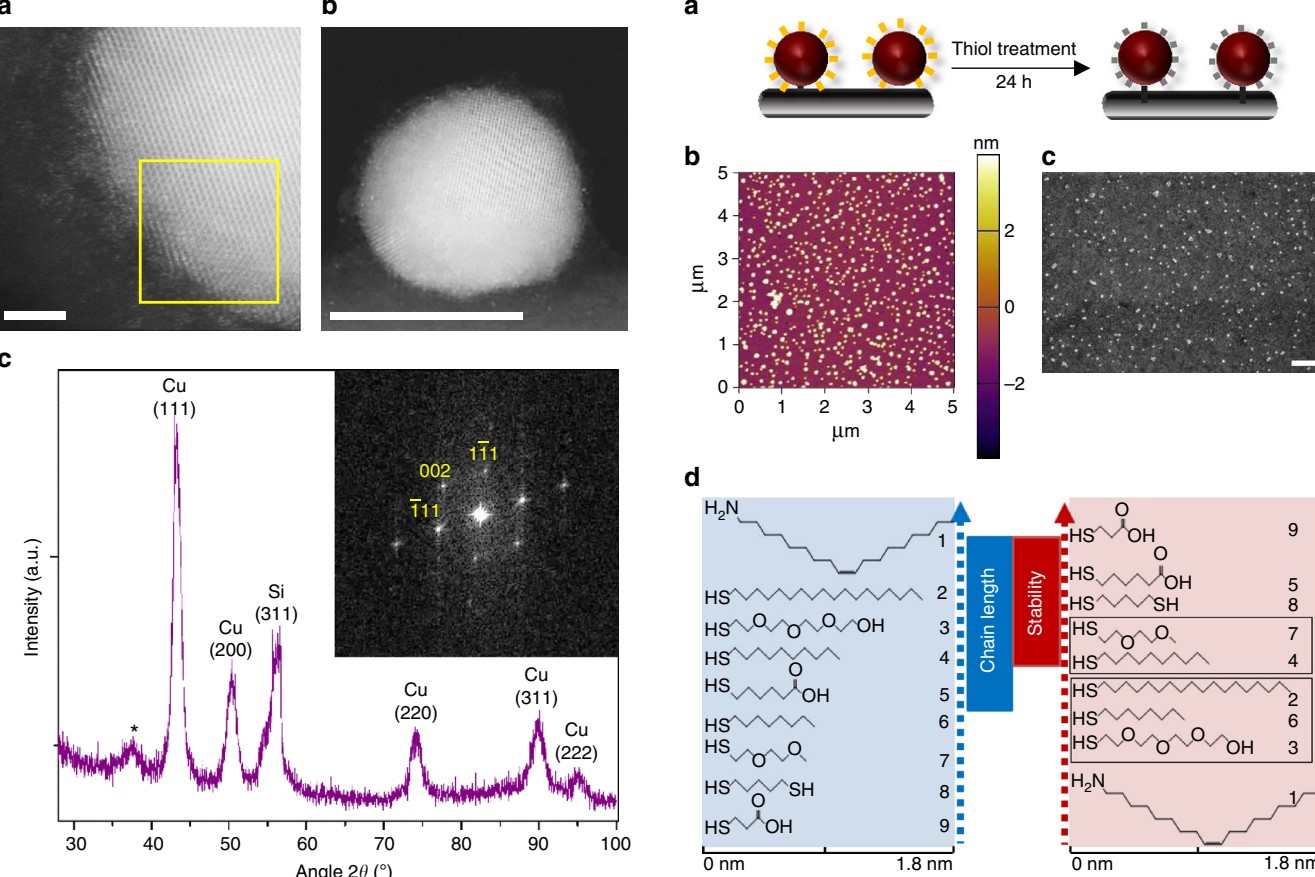

**Fig. 1** ADF-STEM images and XRD spectrum of oleylamine-capped CuNPs. **a** A magnified image of a CuNP showing the crystal lattice and individual Cu atoms surrounding the NP. Scale bar 2 nm. **b** An individual polycrystalline CuNP. Scale bar 10 nm. **c** XRD pattern of oleylamine (1) on Si where the peaks correspond to reflections from Cu (111), (200), (220), (311) and (222). The peak marked with an asterisk ($^*$) is attributed to a minute amount of $Cu_2O$ formed during sample loading into the instrument. (Inset) FFT of the area marked with a yellow square in **a**. The FFT corresponds to the 110 zone axis of Cu. The XRD pattern of our CuNPs perfectly matches the standard ICSD-4349326 data and so it is assumed that all of the CuNPs in this size range have the same fcc (space group: *Fm-3m*) crystal structure.

**Fig. 2** Exchanging oleylamine ligands on tethered CuNPs with thiolate ligands. **a** Schematic representation of the ligand exchange process where oleylamine (1) attached to the CuNPs are replaced with incoming thiol ligands. The thiols are; 1-octadecanethiol (2), 2-{2-[2-(2-mercaptoethoxy) ethoxy]ethoxy}ethanol (3), 1-decanethiol (4), 6-mercaptohexanoic acid (5), 1-octanethiol (6), 2-(2-methoxyethoxy)ethanethiol (7), 1,5-pentanedithiol (8) and 3-mercaptopropionic acid (MPA) (9). The CuNPs are tethered onto the substrate with a monolayer of (3-mercaptopropyl) trimethoxysilane (MPTMS). **b** AFM image (on Si) of tethered 1-octanethiol-capped CuNPs (6), which has undergone soaking in a thiol solution and rinsing with ethanol. **c** SEM image of oleylamine (1)-capped CuNPs on ITO/ Glass. Scale bar 200 nm. **d** Molecular structures of oleylamine and the thiols used for ligand exchange and capping of CuNPs, sorted according to the chain length and stability (with respect to the surface plasmon resonance peak wavelength change) in air when used as a capping layer for CuNPs. The ligands grouped in boxes have similar stabilities

Fig. 2a, which offers the advantage that it allows complete removal of excess ligand by vigorous rinsing after the exchange reaction is complete, and ensures the whole surface of the CuNP is exposed to air when investigating stability towards oxidation in air. Additionally, this approach allows changes in CuNP size as a result of ligand exchange to be probed directly using atomic force microscopy (AFM). The as-synthesised CuNPs are capped with oleylamine, which is easily exchanged with thiolate ligands when the NPs are immersed in a thiol solution, due to the difference in concentration between the oleylamine and thiol species and much greater strength of the S-Cu bond ($\sim$275 kJ mol$^{-1}$ vs. $\sim$160 kJ mol$^{-1}$)[36–38]. The chemisorption of alkylthiols onto Cu has been reported to result in $\sim$90% coverage within 10 s, followed by convergence to a compact monolayer over 24 h[39]. For this reason the ligand exchange reactions were performed over 24 h. Importantly, it has been reported that thiol solutions can etch and/or destabilise the surface of CuNPs even under oxygen and water free conditions[18,40], which would render this class of ligands unusable. To address this point AFM height analysis of thousands of CuNPs tethered to the surface of a silicon wafer was

performed before and after exchange of oleylamine (1) with 1-octanethiol (6) and MPA (9) (Fig. 2b and Supplementary Fig. 4). A polished silicon wafer was the substrate of choice due to its very low surface roughness; 0.3–0.35 nm over an area of $5 \times 5$ μm. Exchange of oleylamine with thiolate was confirmed using X-ray photoelectron spectroscopy (XPS) (Supplementary Fig. 5) which shows disappearance of the peak corresponding to N1s after thiol treatment. Analysis of the data shown in Fig. 3a–c gives the mean particle height of CuNPs capped with ligands 1, 6 and 9 as 6.7 ± 1.4, 5.9 ± 1.1, 7.7 ± 1.9 nm respectively. To a first approximation the decrease in average particle size ($\sim$0.8 nm) upon ligand exchange with 1-octanethiol is consistent with the reduction of the chain length from 18 carbons in oleylamine to 8 in 1-octanethiol (Fig. 2d). However, there is an apparent increase in mean particle size when oleylamine is exchanged with the much shorter MPA (9) ligand. This difference can be understood, in

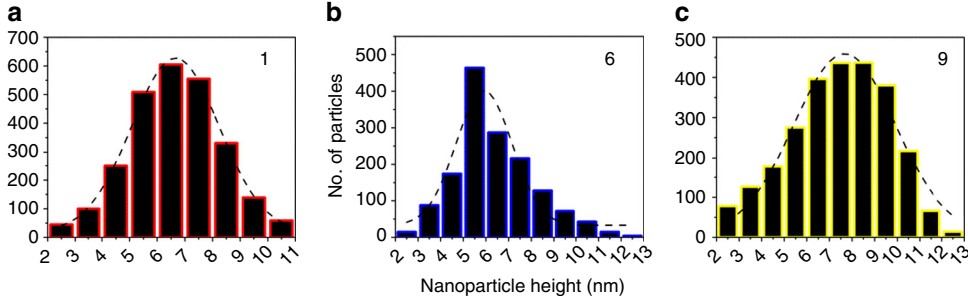

**Fig. 3** AFM particle size analysis of CuNPs with ligands 1, 6 and 9 supported on a Si wafer. **a–c** are respective histograms generated by particle size analysis (using Asylum Research software) for CuNPs capped with ligands 1, 6 and 9. A representative source AFM image for the case of ligand 6, used for particle size analysis, is given in Fig. 2b and AFM images for ligands 1 and 9 are given in Supplementary Fig. 4. Three or four $5 \times 5\,\mu m$ areas of each sample were analysed for this purpose and the standard deviation is given as the ±errors associated with the height measurements

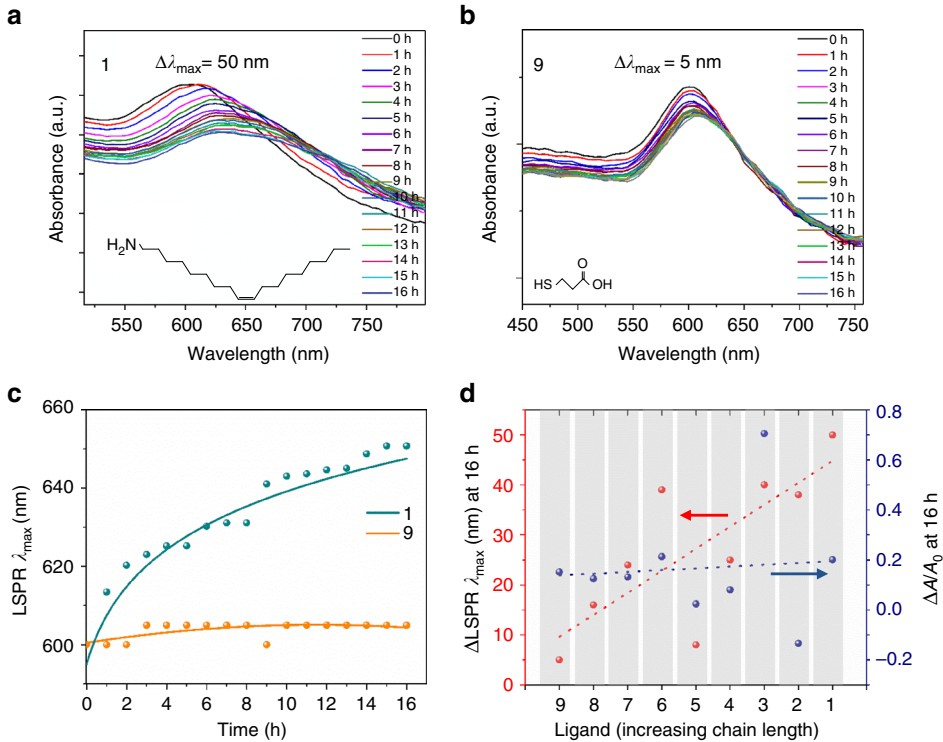

**Fig. 4** UV-vis stability studies of thiol treated CuNPs. **a**, **b** and respective absorbance spectra of 1 and 9 over a period of 16 h in air. The films are sub-monolayers of NPs tethered on glass. A dramatic fall in the shift of maximum wavelength ($\Delta\lambda_{max}$) is observed with thiol treatment. Plot **c** LSPR $\lambda_{max}$ vs time, for samples 1 and 8 shows the stability of MPA-capped CuNPs in air. **d** Shift in $A/A_0$ and LSPR $\lambda_{max}$ from 0 to 16 h of air exposure for various ligand-treated CuNP sub-monolayers

part, in terms of the different nature of the interaction between the AFM tip and the ligand at the NP surface[41], since the alkyl chain of OA is strongly hydrophobic and the carboxylic moiety on MPA is strongly hydrophilic. The size variation can also be attributed to nanoscale differences in the shape of the AFM tip used to image different samples, as has been previously noted by Ebenstein et al.[42] when imaging CdSe nanocrystals, as well as variations in the stiffness/rigidity of the organic capping layer[43]. In our laboratory imaging, the same sample using different AFM tips results in a ±0.6–0.7 nm variation in CuNP height (Supplementary Table 2), which is consistent with the observation of Ebenstein et al.[42].

Given the limitation in the resolution of the AFM measurements, two orthogonal tests for low-level etching of CuNPs by thiol solutions were performed: Firstly, dense multilayer films of oleylamine-capped CuNPs (Supplementary Fig. 6) supported on

1 cm² silicon substrates were soaked in small volumes of the thiols used for AFM height analysis. The solvent was then removed and dried before re-dissolving the residue in nitric acid and testing for Cu content using inductively coupled plasma mass spectroscopy. For 1-octanethiol (6) and MPA (9) the Cu level was <0 ppb and 2.5 ± 0.4 ppb respectively, which is at or below the detection limit of the instrument (see Methods). A value of 2.5 ppb is equivalent to less than one-third of one monolayer of Cu atoms etched from only the top layer of CuNPs in the multilayer film. An equally plausible explanation for the presence of trace Cu in the MPA solution after ligand exchange, is that a small number of MPA-capped CuNPs are dislodged from the multilayer CuNPs film into the solvent during the ligand exchange reaction, since MPA is very soluble in ethanol. Analysis of the solvent used for ligand exchange with 6-mercaptohexanoic acid (5), which is identical to MPA except with two additional –$CH_2$– units in the

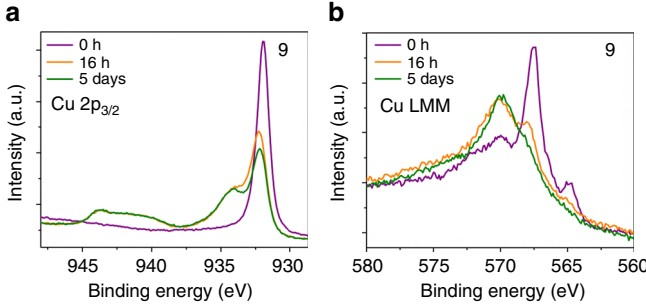

**Fig. 5** Oxidation of CuNPs in air. **a** XPS core level Cu 2p$_{3/2}$ spectra and **b** Cu LMM Auger spectra of sub-monolayers of MPA-capped CuNPs (9) on Au-coated silicon

linker chain, found the Cu level to be <0 ppb, which is consistent with that for the 1-octane thiol and is compelling evidence that low-level etching of CuNPs does not occur. Additionally, samples of the solvent were dried and the thickest areas of solvent residue probed for trace Cu using high-resolution (HR) XPS (Supplementary Fig. 7). The results of that experiment show no evidence for Cu in the solvent used to deposit the thiols 5, 6 or 9 although there is a weak Cu peak in the spectrum from the residue from oleylamine-capped CuNPs soaked in ethanol without thiol, which corroborates the conclusion that trace Cu in the solvent after the ligand exchange reaction does not necessarily result from low-level etching. Consistent with this conclusion it is also noted that scanning electron microscopy (SEM) imaging of samples before and after thiol treatment show no significant reduction in CuNP number density (Supplementary Fig. 8). Taken together the AFM, mass spectroscopy, XPS and SEM data provide compelling evidence that exchange of oleylamine with thiol ligands can be performed without etching the CuNPs.

**Probing CuNP stability towards oxidation in air**. The extinction spectra of a sub-monolayer of CuNPs (Fig. 2c and Supplementary Fig. 8) capped with oleylamine and the thiols shown in Fig. 2d, was measured over a period of 16 h in ambient air (Fig. 4a, b and Supplementary Figs. 9–11). The very large absorption cross-section of plasmon-active CuNPs ensures that even at such low number density the LSPR band is clearly visible. The high sensitivity of the LSPR $\lambda_{max}$ to the local dielectric environment is evident from the large range of starting values for the eight different ligands used, which spans 575–635 nm. The relatively large dielectric constants of Cu$_2$O and CuO (6–8)[6,44,45] also ensures that the LSPR $\lambda_{max}$ is a very sensitive probe of oxide thickness. For CuNPs capped with ligands 1 and 3–9 oxidation in air results in a red-shift of the LSPR $\lambda_{max}$ in conjunction with a decrease in absorbance (A) (Fig. 4c, d)[46]. According to Mie theory, the decrease in LSPR intensity results from a decrease in the size of the metallic core as the surface oxide layer grows[6,17]. Conversely, for CuNPs capped with ligand 2 the absorbance increases with time. According to Ghodselahi et al.[47] a decrease in the dielectric constant of the host matrix increases the absorbance intensity and so the anomalous behaviour for ligand 2 is tentatively attributed to the gradual evaporation of ethanol molecules trapped between the alkyl chains of the ligand, since ethanol is known to be easily trapped in monolayers of dodecane thiol[48,49]. It is evident from Fig. 4d that the change in LSPR $\lambda_{max}$ is a much more sensitive probe of the oxidation stability of metal CuNPs than changes in the absorbance. Whilst studies pertaining to the stability of ligand-capped CuNPs towards air-oxidation are few in number, conventional wisdom is that the most effective passivation of metal nanoparticles is achieved using long alkyl chains that are hydrophobic in nature[50,51]. Counter to this, our results

show that ligands with hydrophilic end groups (carboxylic acid and thiol) are much more effective than hydrophilic alkyl chains. Furthermore, the shortest ligand investigated; MPA (9), imparts the highest stability towards oxidation in air, with only a 5 nm increase in LSPR $\lambda_{max}$ over 16 h (Fig. 4c) increasing to 10 nm after 24 h (Supplementary Fig. 12c), which is remarkable given that the linker chain is only two –CH$_2$– units long. Such a small rate of oxidation represents a dramatic improvement over the best reported in the literature to date[17–19,21] (Supplementary Note 1). The stability of MPA-capped CuNPs is particularly impressive, because the CuNPs are exposed directly to ambient air, whilst in all but one[21] of the aforementioned reports the moisture and O$_2$ must arrive at the CuNPs via the solvent. Notably, after 7 days the change in LSPR $\lambda_{max}$ for MPA-capped CuNPs saturates (Supplementary Fig. 12c), which is compelling evidence that the thickness of the oxide is self-limiting, consistent with the observations of Chen et al.[52] for CuNPs formed by cooling Cu vapour.

Analysis of the Cu:S ratio determined from the XPS data (Supplementary Table 3) shows that for the three alkylthiols the ligand packing density increases with increasing alkyl chain length, consistent with the expectation that a longer alkyl chain forms a more dense capping layer due to attractive inter-chain van der Waals interactions[17,18,51]. However, there is no trend in the stability towards air oxidation, since translating from 1-octanethiol (6) to 1-decanethiol (4) the resistance to oxidation increases, but then decreases for octadecanethiol (2). Of all the thiol ligands investigated, 2-{2-[2-(2-mercaptoethoxy)ethoxy]ethoxy}ethanol (3) proved to be least effective at slowing air oxidation, which is consistent with the finding that it has the lowest ligand packing density, combined with the disorder in ether ligand layers that is known to result from the repulsive interaction between oxygen atoms in adjacent ether chains[50], which together render the ligand layer more permeable to oxygen ingress. Comparison of the ligand packing density for thiols with ether chains (3 and 7) and carboxylic acid terminated alkyl chains (5 and 9) reveals that, in a reversal of the trend for the simple alkyl thiols, shorter ligands are more densely packed for these classes of ligand, which correlates with improved stability towards air oxidation. Strikingly, alkyl thiol ligands terminated with hydrophilic end groups-COOH (or -SH) offer the highest stability overall. The importance of the end group functionality in determining the effectiveness of the ligand towards blocking oxidation of CuNPs is highlighted by the fact that the stability of CuNPs capped with MPA and (5) is similarly very high, although there is a significant difference in the thiol packing density between these two ligands.

To determine the changes in the chemical state of the CuNPs upon exposure to air, XPS core level Cu 2p$_{3/2}$ spectra and Cu LMM Auger spectra of MPA-capped CuNPs (9) were collected after 0, 16 h and 5 days of air exposure (Fig. 5 and Supplementary Fig. 13). The narrow peak at 932.0 eV[53] in the Cu 2p$_{3/2}$ XPS spectrum is consistent with Cu$^0$, which is confirmed by the resemblance of the Cu LMM Auger spectrum (Supplementary Fig. 13d) to that of pure Cu[53]. After 16 h in air, the CuNPs are partially oxidised, which is evidenced by the satellite peaks in the XPS spectrum between 940.2 and 943.8 eV and the shoulder peaks at 933.9 and 934.9 eV corresponding to CuO (Fig. 6a). The peak at 932.3 eV can be fitted with two peaks, corresponding to Cu$^0$ at 932.1 eV and Cu$_2$O at 932.4 eV. The new peak at 570.0 eV in the Auger spectrum is assigned to Cu$_2$O[53]. The sample exposed to air for 5 days exhibits more pronounced peaks at 933.9 and 934.9 eV (Fig. 5a) due to the growth of CuO and a further reduced peak at 567.9 eV in the Auger spectrum (Fig. 5b) due to depletion of metallic Cu over time. Therefore, analogous to the oxidation behaviour of bulk copper and thin films[54,55], the formation of Cu$_2$O initiates during a short period of air

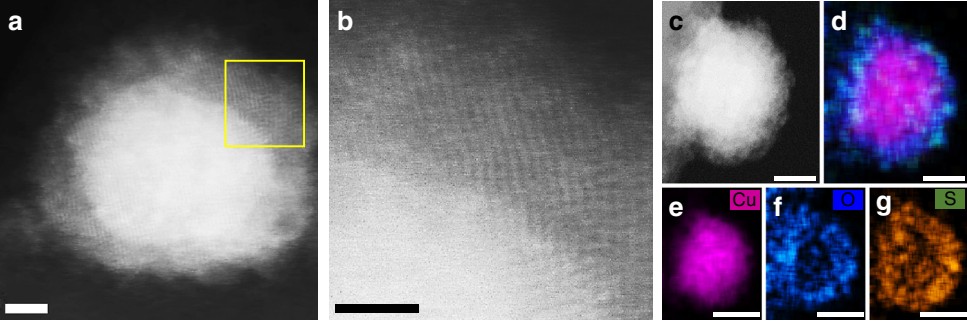

**Fig. 6** Thickness of copper oxide shell. **a** ADF-STEM image of a MPA-capped CuNP exposed to air for 10 days displaying the oxide core shell around the CuNP. Scale bar 2 nm. **b** A magnified region (marked in yellow in **a**) of the highly crystalline core shell. Scale bar 2 nm. **c–g** EDX elemental scans of a similar particle confirming the presence of an oxide core shell. The appearance of S is from the ligand MPA. Scale bar 5 nm (**c**, **d**). Scale bar 10 nm (**e–g**)

exposure[6,12]. Interestingly, our results show that CuO starts to form within 16 h of air exposure, which is a shorter time scale than reported for thin Cu films: Platzman et al.[53] have reported that for thin films of polycrystalline Cu, within 24 h of air exposure CuO forms, whereas Ijima et al.[56] reports a 13-day induction time prior to CuO formation. CuNPs have a much higher surface/volume ratio than thin films, and so it is reasonable that the rate of formation of CuO is greater[18]. The thickness of each oxide layer is estimated using the thickogram method[57] (Supplementary Note 2) assuming a topographic correction factor of 0.67—applied to account for small particle size and spherical shape[58,59]. The thickness of $Cu_2O$ is ~1 nm after both 16 h and 5 days of air exposure, and the thickness of the CuO layer increases only slightly from ~1.1 to ~1.3 nm between 16 h and 5 days within a ±10% error. This saturation in oxide thickness is consistent with the evidence for self-limiting oxide thickness provided by the extinction study. Corroborating these findings, the oxide layer thickness after 10 days air exposure was directly determined using ADF-STEM imaging; Fig. 6a and b (and Supplementary Fig. 14), and found to vary between 2 and 2.7 nm. The crystalline core shell is confirmed as copper oxide using energy-dispersive X-ray spectroscopy (EDX) elemental scans (Fig. 6bc–g and Supplementary Fig. 14).

**Work function**. How the work function ($\varphi$) of small metal NPs scales with size is critically important for the advancement of molecular electronics (where NPs are used as electrodes for single-molecule devices), NP-semiconductor composites for the emerging generation of thin film photovoltaics and plasmonic hot-electron devices because the $\varphi$ is a critical determinant of the energetics at the NP-semiconductor interface. Remarkably there have been only seven studies addressing this important point[22–28] and there is no experimental consensus as to how $\varphi$ correlates with NP size. A classical treatment for spherical metal particles large enough not to exhibit semiconducting character due to quantum confinement effects (>2–3 nm)[60,61], predicts that the $\varphi$ should decrease with increasing NP diameter, converging to the bulk value for diameters ≥ 22 nm[62]. Experimental evidence supporting this model has been provided by Zhou et al.[22], Schmidt-Ott et al.[23] and Müller et al.[24] for naked Ag NPs in the gas phase using differential mobility analysis and photoelectric studies, respectively. Using the Kelvin probe technique Schnippering et al.[25] have corroborated these findings for citrate-capped Ag NPs. Conversely, it has also been reported that the opposite trend occurs: Dadlani et al.[26] for Ag NPs capped with polyvinylpyrrolidone and dodecanethiol based on measurements of NP $\varphi$ using ultraviolet photoelectron spectroscopy (UPS), which

is attributed to the nature of the ligand. Using KPFM, Zhang et al.[28] have also reported that the $\varphi$ of ligand-capped Au increases with nanoparticle diameter, which they propose results from the interaction with the underlying substrate.

The very slow rate of oxidation of MPA-capped CuNPs in air and NP size distribution of particles synthesised as part of this work; 3–17 nm, presents an ideal opportunity to determine how the $\varphi$ of small metallic particles scales with NP diameter. To date, how the $\varphi$ of isolated CuNPs scales with size has not (to our knowledge) been reported, although it can be assumed that the same physics applies to all spherical metal particles regardless of the element[60]. The technique used in the current study is KPFM because it allows for the simultaneous acquisition of NP diameter (height) to an accuracy of ~0.7 nm (limited by the roughness of the supporting substrate and the shape of the AFM tip), and the contact potential difference (CPD) for a large number of NPs within a few minutes (i.e. the time taken to collect an image), so statistically significant data sets can be acquired rapidly. Conversely it is extremely costly to synthesise bulk samples of NPs with the very tight size distribution needed to reliably investigate the correlation between NP $\varphi$ and diameter using UPS and KP measurements, which yield the lowest $\varphi$ and average $\varphi$ of a large sample respectively.

In the first instance, NPs were tethered to a heavily *n*-doped Si substrate using a monolayer of MPTMS (Fig. 7a, inset). The density of NPs on the substrate was chosen to ensure minimal clustering and that, on average, the NPs are separated by a distance much greater than the AFM tip radius of ~20 nm. Only isolated NPs with a circular footprint were included in the analysis (Fig. 7b). Determination of the absolute magnitude of NP $\varphi$ is difficult because it requires knowledge of the $\varphi$ of the AFM tip, which is extremely difficult to determine with a high degree of accuracy. Consequently, care must be taken to determine whether the measured correlation of CPD with NP size actually corresponds to an increase or decrease in $\varphi$ with NP size. To determine this, a thin film of Al was thermally evaporated onto a Si wafer followed by a thin layer of Au. This bilayer film was then scored to expose the underlying Al and the variation in CPD across the score measured using the KPFM (Supplementary Fig. 15). It is well established that Au has a much higher $\varphi$ than $Al/Al_2O_3$ (~5.1 eV vs. ~3.9 eV, respectively), even when exposed to the ambient air, and so by comparing the direction of change with the trend observed for MPA-capped CuNPs on MPTMS functionalised Si (Fig. 7c), it is confirmed that the $\varphi$ of the CuNPs decreases with increasing NP diameter (Fig. 7a and Supplementary Fig. 16). To corroborate this conclusion, the $\varphi$ of a sub-monolayer of MPA-capped CuNPs tethered on to a Si substrate with a monolayer of MPTMS (Fig. 7b) was measured using a

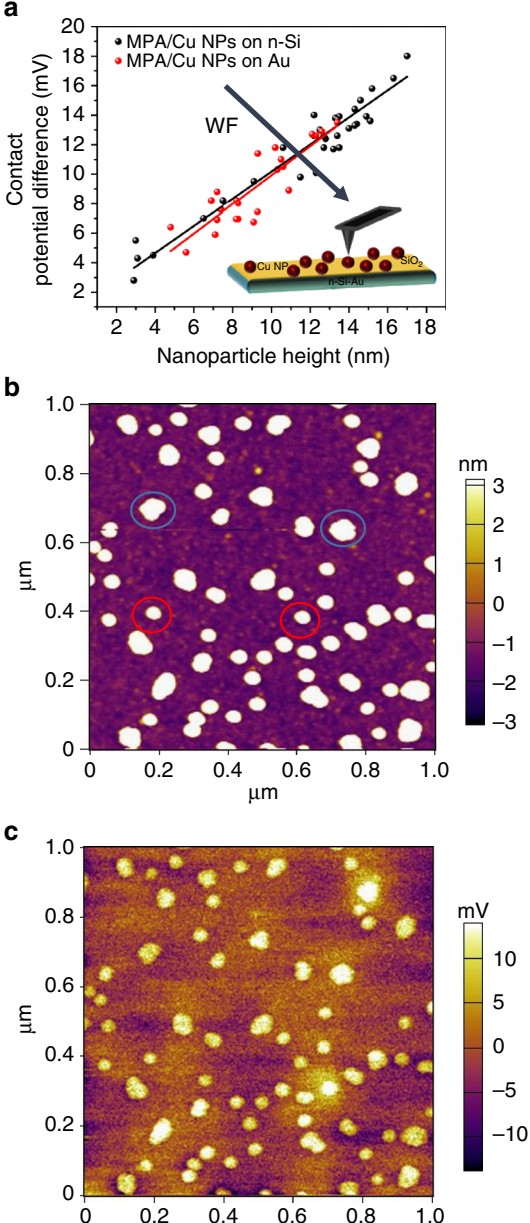

**Fig. 7** KPFM analysis of CuNPs-capped with MPA. **a** Contact potential difference vs. NP height for MPA-capped CuNPs on n-Si and n-Si/Au. The error in CuNP height measurement for all data points is estimated to be ±0.7 nm. (Inset) Schematic representation of the sample of MPA-capped CuNPs used for KPFM analysis where the particles are tethered onto a surface of SiO₂/n-Si using a monolayer of MPTMS. **b** AFM height image of MPA-capped CuNPs on Si. The particles marked with a blue circle are examples of aggregates of NPs and those with a red circle are individual NPs that have been used for the analysis. **c** The contact potential difference image of the corresponding image in **a**

Kelvin probe to be $4.21 \pm 0.02$ eV (the values reported in the literature for bulk Cu vary between 4.2 and 4.6 eV under N₂ and ambient conditions[63]), and compared to that of a reference Si substrate with a monolayer of MPTMS treated with the MPA solution used for ligand exchange and washed with toluene in order to ensure that it undergoes the same treatment as the sample with CuNPs. The $\varphi$ of the reference sample was $4.30 \pm 0.01$ eV. Since the reference Si substrate has a slightly higher $\varphi$ than with a sub-monolayer of CuNPs, the $\varphi$ of the MPA-capped

CuNPs must be lower than that of MPTMS functionalised Si because the KP measures the average $\varphi$ under the probe. Together the results of this experiment show that the $\varphi$ of the MPA-capped CuNPs decreases with increasing NP diameter.

Importantly, the CPD method requires that the Fermi level of the sample and probe are aligned. For this study, n-doped Si was the substrate of choice because of its very low surface roughness, although this substrate also has a very thin and defective native oxide layer. In principle, the native surface oxide layer combined with the MPTMS layer (thickness 0.7–0.8 m) are sufficiently thin and defective for thermodynamic equilibrium to be established between the CuNPs and Si. To confirm this, the same set of measurements were also performed for MPA-capped NPs tethered to a Au electrode modified with 1,5-pentanedithiol (length ~1 nm). It is evident from Fig. 7a that the same trend is observed for the variation of $\varphi$ with the NP size: The CuNP $\varphi$ decreases with increasing NP diameter consistent with the classical relationship of $\varphi$ and particle size (for single-charged species) based on classical image and Coulombic potentials of spherical geometry[62].

The aforementioned analysis assumes that the packing density of ligands at the CuNP surface does not change over the range of NP sizes investigated, which has been reported to be the case in a few instances for ligand-capped Au NPs[64]. However, more typically the density of ligands at the surface of spherical metal NPs is expected to decrease with increasing NP diameter due to the radius of curvature[64,65]. Given that the change in NP $\varphi$ that results from modification of the NP surface potential by the ligand layer is expected to scale with molecule density[63,66], any significant decrease in MPA ligand density would complicate the interpretation of the data in Fig. 7. For MPA, the inward pointing dipole associated with the thiolate linkage to Cu[67] and electron withdrawing nature of the carboxylic acid end group would both operate to increase the surface potential contribution to the NP $\varphi$. Consequently, a reduction in MPA packing density with increasing NP diameter could give rise to the observed decrease in NP $\varphi$ with increasing diameter. To investigate this possibility, the CPD of oleylamine-capped CuNPs on Si was measured, taking special care to ensure the sample was analysed quickly after its removal from the glovebox (Supplementary Fig. 17). Although oleylamine-capped CuNPs are much less stable than MPA-capped CuNPs, the rate of oxidation (Supplementary Fig. 9a) is much slower than the time taken to collect the KPFM image. In contrast to the case of MPA, the polarity of the Cu-primary amine bond[68] together with the electron-donating character of the alkyl chain[63] operate to reduce the $\varphi$ of the CuNP from its bare value, and so a reduction in ligand density with increasing NP diameter would result in an increase in the $\varphi$ of oleylamine-capped CuNPs. It is evident from the data in Supplementary Fig. 17 that the direction and magnitude of the change in $\varphi$ is in fact very similar to that measured for MPA-capped CuNPs, and so it can be concluded that the variation in CPD with NP diameter results from a size effect, rather than a variation in ligand density. Interestingly the magnitude of the change of $\varphi$ with size for our substrate-bound CuNPs is also of the same order of magnitude as reported by Zhou et al.[22] for free Ag NPs in air, where the variation in $\varphi$ for NPs in the same size range is comparable.

## Discussion

We have shown that alkylamine ligands used in the synthesis of CuNPs can be readily exchanged with thiolate ligands without etching the Cu core, and that very short alkyl thiols with a carboxylic acid end group are exceptionally effective at retarding oxidation of CuNPs in air. These finding opens the door to the

possibility of using CuNPs in emerging electronic devices in place of costly Ag and Au NPs, because the susceptibility to oxidation is dramatically reduced using a capping layer thin enough not to electrically isolate the metal from its surroundings. We have also addressed the important fundamental question of how the work function of small supported metal particles scales with size, a relation that has until now has been the subject of debate. Although it is not possible to know the exact magnitude of the work function of nanoparticles using KPFM, due to the size and shape of the AFM tip, it is notable that the magnitude of the change in $\varphi$ with CuNP diameter over the size range 3–17 nm is of the same order as that reported for unsupported Ag particles[22]. A variation of this magnitude is comparable to the thermal energy of an electron at room temperature (∼25 meV) and so for practical electronic applications such a size dependence of the $\varphi$ of metal NPs would not be an important determinant of the energetics at an interface with a semiconductor. Taken together these finds set the stage for a more intense research effort into the utility of CuNPs for emerging electronic applications, including the development of hybrid electronic materials based on colloidal metal nanoparticles and organic/perovskite/transition metal oxide semiconductors in which the CuNP is strongly electrically coupled to the surrounding semiconductor.

## Methods

**Materials**. All materials were purchased from Sigma-Aldrich unless otherwise stated. The materials and solvents were used as received without further purification.

**CuNP synthesis**. The process of NP synthesis is a modification of the method reported by Grouchko et al.[29]. A mass of 40.00 mg of copper acetylacetonate (Cu (acac)$_2$) and 40 ml of oleylamine was added to a pressure round bottom flask (with a thermowell) and was sealed inside a N$_2$-filled glovebox. The reaction mixture was then stirred at 40 °C on a heating block for 30 min until the Cu(acac)$_2$ was fully dissolved resulting in a green-coloured solution. The temperature was increased at a rate of 2 °C per min until a temperature of 170 °C (thermowell) was achieved. The colour change was green (until 100 °C), yellow (110 °C), orangish-green (130 °C), khaki (150 °C), brown (160 °C) and reddish-brown (170 °C). The solution was stirred at 170 °C overnight (17 h). The final solution was allowed to cool to room temperature where upon a reddish-brown precipitate was visible on the bottom and walls of the flask. The flask was transferred to a glovebox and the contents were centrifuged for 10 min at 8000 rpm. The supernatant (excess oleylamine—yellow) was discarded, and 3 ml of anhydrous ethanol was added to the precipitate (reddish-brown). The solution was centrifuged for 10 min at 8000 rpm and the supernatant containing excess oleylamine in ethanol was discarded. This process was repeated and the final precipitate was dispersed in 10 ml of toluene to form a well-dispersed CuNP solution bright reddish-brown in colour.

**Monolayer/thin film preparation for ligand exchange**. The substrates (silicon, glass or indium tin oxide (ITO) coated glass) were exposed to 95% (3- mercaptopropyl)trimethoxysilane (MPTMS) vapour for 2 h for a monolayer of MPTMS to form on the surface of the substrate. A four times diluted solution of the CuNP stock solution in toluene was then spun on glass, ITO/Glass or Si at 500 rpm for 60 s followed by a second step at 2000 rpm for 3 s. The films were annealed at 70 °C for 10 min to evaporate the remaining toluene. The MPTMS monolayer deposition was performed in air and the deposition of CuNP films and drying was done inside a N$_2$-filled glovebox.

**Ligand exchange**. CuNP films were submerged in $5 \times 10^{-2}$ M thiol solutions for 24 h, in a nitrogen-filled glove box. Such a long exposure time was used to ensure that sufficient time had been given for complete ligand exchange[39]. The thiol-treated substrates were then washed with anhydrous ethanol prior to annealing at 50 °C for 5 min to get rid of remaining ethanol.

**Electronic absorption spectroscopy**. Cary 60 UV-Vis and Cary IE UV-Vis (Agilent Technologies) were used for extinction measurements. Sub-monolayers of CuNPs were prepared from a four times diluted solution of CuNPs on glass or quartz using the deposition method described above. The measurements were carried out under ambient conditions.

**X-ray photoelectron spectroscopy**. The samples for X-ray photoelectron spectroscopy were prepared on thin layers of Au (30 nm) deposited on Si substrates coated with a mixed monolayer of MPTMS:APTMS (3-aminopropyl)

trimethoxysilane). A four times diluted solution of oleylamine-capped CuNPs was then spun on the substrate as a two-step process: 500 rpm for 1 min and 2000 rpm for 3 s prior to annealing at 70 °C for 10 min. The resulting films were soaked in various thiol solutions ($5 \times 10^{-2}$ M) for 24 h for ligand exchange to proceed. The films were then rinsed with anhydrous ethanol to remove the excess MPA and dried at 50 °C for 5 min. XPS measurements were performed using a Kratos Axis Ultra DLD spectrometer. The samples were illuminated using X-rays from a monochromated Al Kα source (hν = 1486.6 eV) and detected at a take-off angle of 90°. The resolution, binding energy referencing and transmission function of the analyser were determined using a clean polycrystalline Ag foil. XPS peak fitting was carried out using the CasaXPS software (Voigt -mixed Gaussian–Lorentzian line shapes and a Shirley background). The peaks were corrected with respect to C1s at 284.7 eV due to the use of neutraliser to avoid charging.

**ADF-STEM**. The samples were prepared on lacey carbon 200 mesh square nickel grids from EM Resolutions. Samples for imaging of oleylamine-capped CuNPs were prepared by adding a drop of CuNPs in toluene and drying at 70 °C for 10 min. Samples to measure the formation of copper oxides were prepared as follows: a lacey carbon 200 mesh square nickel grid was exposed to MPTMS vapour in vacuum for 2 h. The TEM grid was then coated with a drop of oleylamine-capped CuNPs inside a N$_2$ filled glovebox and heated at 70 °C for 10 min. The grid was soaked in $5 \times 10^{-2}$ M MPA followed by washing with anhydrous ethanol and drying at 50 °C for 5 min. The grid was exposed to ambient air for 10 days. The structure and morphology of the NPs were analysed using a doubly corrected scanning transmission electron microscope, ARM200F (200 kV). ADF-STEM images were recorded using a JEOL annular dark field detector. EDX data were collected with and Oxford Instrument X-Max extreme windowless 150 mm$^2$ SDD detector.

**AFM and KPFM**. Imaging was performed using an Asylum Research MFP3D-SA operated in AC (tapping) mode. The samples were prepared on n-doped Si substrates functionalised with a monolayer of MPTMS prior to the deposition of CuNPs. Image analysis was performed using the in-built particle analysis option of Asylum Research software, which generates histograms of particle size distribution. The raw images were flattened using the magic mask option with a flatten order of 1.

**SEM**. Images were taken using a Zeiss Supra 55-VP FEGSEM operating at 10 kV with the Inlens detector. The samples were prepared on n-doped Si or ITO/Glass substrates functionalised with a monolayer of MPTMS prior to the deposition of CuNPs.

**Kelvin probe**. Bulk work function measurements were performed using a Kelvin probe referenced to freshly cleaved highly oriented pyrolytic graphite (HOPG) in a N$_2$-filled glovebox.

**X-ray diffraction**. Thick films of CuNPs on Si were prepared using a drop cast method. X-ray diffraction (XRD) measurements were carried out using a Siemens D5000 X-ray diffractometer operated in grazing angle using Cu (Kα) radiation with a wavelength of 1.542 Å. The scans were run for 14 h (for better peak resolution) and the samples were mounted inside a purged graphite dome under a continuous flow of N$_2$ to avoid formation of copper oxides during prolonged scans. The XRD peaks were assigned using the software Mercury (2016 CSDS release) and the CDS National Chemical Database.

**Inductively coupled plasma mass spectroscopy**. *Instrument*. Agilent 7500cx. Analysis conditions: plasma gas 15 L per min, aux gas 0.9 L per min, make-up gas 0.15 L per min, RF power 1550 W, RF matching 1.8 V, analogue HV 1750V, Pulse HV 1130 V, spray chamber temperature 15 °C, nebuliser pump 0.08 rps. Internal standard (50 ppb Er solution), is introduced into the sample flow through T-pieces. Before acquisition: uptake speed 0.3 rps, uptake time 30 s and stabilisation time 30 s.

*After acquisition*. Probe is rinsed on rinse port with water for 10 s, then 30 s with 3% nitric acid at 0.2 rps, followed by water for 80 s at 0.15 rps. All reading was done in triplicate. 5 mL of 4% ultrapure nitric acid were added to the Teflon round bottom flasks containing solvent residue and left for 5 h. Solution transferred to a 15 mL falcon tube and analysed. Calibration range from 1–1000 ppb.

**Data availability**. All data supporting this study are provided as supplementary information accompanying this paper.

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

## Acknowledgements

We thank Dr D. Walker and Dave Hammond for the assistance with setting up XRD measurements, Dr Neil Wilson for training and providing AFM facilities for KPFM measurements, Dr Nikola Chmel for providing training to use the Cary IE UV-Vis, Dr Jaemin Lee, Kenny Marshall and Vanessa Kavuma for the assistance in film preparation and deposition. Dr Lijiang Song and Philip Aston for performing the ICPMS analysis. We also thank the Leverhulme Trust for the award of research grant (RPG-2015-044) and the United Kingdom Engineering and Physical Sciences Research Council (EPSRC) for funding (grant number: EP/N009096/1).

## Author contributions

G.D.M.R.D. synthesised the CuNPs, carried out ligand exchange reactions, performed SEM, AFM, XRD, UV-Vis, KPFM measurements, analysed the data and wrote the manuscript. M.W. performed the XPS measurements and assisted with the analysis. H.J.P. deposited the NP films for various experiments. A.M.S. and R.B. carried out the HR-TEM measurements and helped with TEM analysis. R.A.H. conceived the study, analysed the data and wrote the manuscript.

## Additional information

**Competing interests:** The authors declare no competing financial interests.

