## [Peer Review File · Nature Communications]

Reviewers' comments:

Reviewer #1 (Remarks to the Author):

The work by Hatton et al., describes the evaluation of ligated copper nanoparticles' susceptibility to oxidation after ligand exchange with thiols of different chain lengths and hydrophobicity. The mechanism for thiol attachment and transformation of the surface in the presence of excess thiol ligands during ligand exchange was reported based on changes in height using AFM. Evaluation of the work function of the MPA protected CuNPs was also reported and ϕ was reported to decrease with increasing size. The reported work seems to be done with care, but this reviewer has questions regarding the presentation and interpretation of the results.

Questions/Comments:

Seemingly inherent in the manuscript, the authors assume the passivation of the NP surface is uniform in both coverage and electronic structure, because no mention of surface availability or binding to the surface is discussed nor any measurements to examine these factors.

- The XPS results for only 3 ligands are presented in Fig. S2 and there is shift in the binding energies for the Cu 2p and Cu LMM regions with different thiols. Is this due to charging or does the Auger parameter remain the same? What happens in ligands with ether bonds (i.e. is there a change in the electron density of the sulfur?) Also, what samples were examined with XPS?
- Why are the relative ratios of Cu and S not reported for the samples examined with XPS?
- Only the 9-MPA N (1s) region is presented in Fig. S4 which describes two thiol capped (1-octanethiol). Where is this data? All of the XPS data sets should be available and omission of the data should be addressed.
- There is no direct evidence provided to support the claim that thiols are not etching the nanoparticle surfaces. The details for image analysis and an orthogonal method to examine the mass distribution (e.g., elemental analysis) are necessary to substantiate this claim.
- Number density is not sufficient to examine etching processes. What is in the washed thiol solution after removal? Again, the data is insufficient to support the authors' conclusion, because it is plausible to assume uncertainty from tip contributions could be similar to the mass loss due to etching if occurred.
- Although not a requirement for the current work, the authors posit reasons for observed stability based on optical properties. However, again, this highlights the need to better characterize the surface of the NPs (e.g., coverage). On page 12-13, the authors provide observational differences based on chain length or terminus. However, these speculative claims could be directly investigated with spectroscopic methods, including XPS. These should be included in the work to augment the impact to the field for further advancing design principles.
- In regards to ϕ measurements, although the most stable NP was examined, how does the ligand passivation of the surface contribute to measurement (i.e., what is the predicted behavior if the packing density was changed, and thus, changed the dielectric environment as was demonstrated in thin films with Au (Phys. Chem. Chem. Phys. 2016, 3675)?
- Would the authors predict based on the current system to have ϕ on the same order of magnitude with bare particles as reported in their comparison with Zhou et al?
- A minor point, size change resulting from thiols exchanging oleylamine on the NP surface is unlikely to be simply described as 0.4 nm change in radial thickness (packing density and attachment) is only qualitative. Uncertainty of some kind should be reported from AFM measurements, because it is similar to the reported uncertainty due to tip effects (Table S1). This reviewer highlights this point because it represents the issues with, in this reviewer's opinion, the unsubstantiated claims outline above.

Reviewer #2 (Remarks to the Author):

The paper reports a fascinating finding on the stability of copper nanoparticles when coated with organic thiols. Interestingly the authors observe not only good stability of the particles in thiol solution but also that short chain acid thiols protected the copper against oxidation more than long chain thiols. Having prepared the copper nanoparticles they have considered the variation in work function with particle size. Whilst I enjoyed the paper and felt it had general interest I am of the opinion that two points require clarification prior to acceptance:

1. In looking at the dissolution of copper in thiol containing solutions the solvent should have been tested for trace copper to prove no dissolution had occurred.
2. Given the range of sizes of copper nanoparticles in the system and assuming uniform slow oxidation of the samples I would expect the spectra in figure 4 to change shape with time, an explanation as to why this does not occur should be provided.

Two minor points:

1. Line 226 - missing unit on 11.7
2. Line 279 - remove at

Response to Reviewers' comments:

Reviewer #1 (Remarks to the Author):

The work by Hatton et al., describes the evaluation of ligated copper nanoparticles' susceptibility to oxidation after ligand exchange with thiols of different chain lengths and hydrophobicity. The mechanism for thiol attachment and transformation of the surface in the presence of excess thiol ligands during ligand exchange was reported based on changes in height using AFM. Evaluation of the work function of the MPA protected CuNPs was also reported and ϕ was reported to decrease with increasing size. The reported work seems to be done with care, but this reviewer has questions regarding the presentation and interpretation of the results.

Questions/Comments:

Point 1: Seemingly inherent in the manuscript, the authors assume the passivation of the NP surface is uniform in both coverage and electronic structure, because no mention of surface availability or binding to the surface is discussed nor any measurements to examine these factors.

Response 1: We would like to thank Reviewer 1 for taking the time to so carefully review our manuscript and for his/her very constructive remarks. We have now included XPS data for all of the ligands investigated (Supplementary Information 5), including analysis of the Cu:S ratios which enables comparison of the ligand density at the nanoparticle surface (Supplementary Table 3). This additional information is discussed on pages 13 and 14 of the revised manuscript, which strengthens

and supports our original conclusions. The XPS data also confirm the facile nature of the exchange of the weakly bound primary amine used in nanoparticle synthesis with the strongly bound thiol ligands.

Point 2: (i) The XPS results for only 3 ligands are presented in Fig. S2 and there is shift in the binding energies for the Cu 2p and Cu LMM regions with different thiols. **(ii)** Is this due to charging or does the Auger parameter remain the same? **(iii)** What happens in ligands with ether bonds (i.e. is there a change in the electron density of the sulfur?) **(iv)** Also, what samples were examined with XPS?

Response 2:

(i) and **(iv)** The XPS results for all 9 ligands investigated have now been included in the Supplementary Information: Supplementary Figures 5 and Table 3.

(ii) The Auger parameter is the same within error and so the apparent shift in binding energy is attributed to an instrumental effect. To confirm this, we have repeated the experiment, making all measurements on the same day, and there are no significant differences in binding energies for either the Cu 2p or Cu LMM regions. This new data is included in the Supplementary Information (Figure 3) alongside the original data.

(iii) We have now added a table summarizing the Auger parameters for all ligand types (including both ethers) to the Supplementary Information; Table 1, from which it is evident there is no correlation between ligand type and the Auger parameter (i.e. no change in chemical environment with ligand type).

Point 3: Why are the relative ratios of Cu and S not reported for the samples examined with XPS?

Response 3: The Cu:S ratios for all of the ligands investigated have now been included in the Supplementary Information and are discussed on page 13-14 (highlighted in red).

Point 4: Only the 9-MPA N (1s) region is presented in Fig. S4 which describes two thiol capped (1-octanethiol). Where is this data? All of the XPS data sets should be available and omission of the data should be addressed.

Response 4: We have now included all of the XPS data for all of the ligands: Supplementary Figure 5. In the original manuscript we only provided the data needed to prove that primary amine ligand (i.e. the ligand used in the copper nanoparticle synthesis) can be exchanged with thiol. Only the N1s and S2p regions were shown in the original manuscript because oleylamine contains N but no S, and the thiol ligands contains S but no N.

Point 5: (i) There is no direct evidence provided to support the claim that thiols are not etching the nanoparticle surfaces. **(ii)** The details for image analysis and an orthogonal method to examine the mass distribution (e.g., elemental analysis) are necessary to substantiate this claim.

Response 5:

(i) We thank Reviewer 1 for raising this important point. We have now directly tested the solvent residue for the two thiol ligands on which AFM height analysis was performed (plus one other to confirm our conclusions) using both inductively coupled plasma mass spectroscopy and high resolution photoelectron spectroscopy. Collectively the findings of these additional experiments support our original hypothesis that primary amines can be exchanged with thiols without eroding

the Cu nanoparticle. These additional experiments are discussed on page 10 and 11 of the revised manuscript (highlighted in red) and the data is given in Supplementary Figure 7.

(ii) Further details of the AFM image analysis have now been included in the experimental section: page 26 (highlighted in red).

Point 6: Number density is not sufficient to examine etching processes. What is in the washed thiol solution after removal? Again, the data is insufficient to support the authors' conclusion, because it is plausible to assume uncertainty from tip contributions could be similar to the mass loss due to etching if occurred.

Response 6: As requested we have now examined the copper content of thiol solution after ligand exchange and find that there is an insignificant level of copper. Please see response to Point 5.

Point 7: Although not a requirement for the current work, the authors posit reasons for observed stability based on optical properties. However, again, this highlights the need to better characterize the surface of the NPs (e.g., coverage). On page 12-13, the authors provide observational differences based on chain length or terminus. However, these speculative claims could be directly investigated with spectroscopic methods, including XPS. These should be included in the work to augment the impact to the field for further advancing design principles.

Response 7: We thank Reviewer 1 for raising this point. As suggested we have now used our XPS analysis to determine the Cu:S ratio for all of the ligands Supplementary Table 3, from which it is possible to draw conclusions about the relative thiol packing density, which adds further depth to the discussion on page 13-14 (highlighted in red). We have also now added a table summarizing the Auger parameters for all ligand types (including both ethers) to the Supplementary Information; Table 1, from which it is evident there is no correlation between ligand type and the Auger parameter (i.e. no change in chemical environment with ligand type).

Point 8: In regards to ϕ measurements, although the most stable NP was examined, how does the ligand passivation of the surface contribute to measurement (i.e., what is the predicted behavior if the packing density was changed, and thus, changed the dielectric environment as was demonstrated in thin films with Au (Phys. Chem. Chem. Phys. 2016, 3675)?

Response 8: We thank Reviewer 1 for making this point, which we have now addressed by performing an additional experiment (Supplementary Information Figure 17) the findings of which are described in the following paragraph which has been added to the manuscript on pages 19 and 20, as well as the reference suggested.

The aforementioned analysis assumes that the packing density of ligands at the Cu NP surface does not change with NP diameter, which has been reported to be the case for MPA capped Au NPs over the same range of NP diameters.⁷⁰ However, more typically the density of ligands at the surface of spherical metal NPs is expected to decrease with increasing NP diameter due to the radius of curvature.⁷¹ Given that the change in NP ϕ that results from modification of the NP surface potential by the ligand layer is expected to scale with molecule density^{72,73} any significant decrease in MPA ligand density would complicate the interpretation of the data in Figure 8. For MPA the inward pointing dipole associated with the thiolate linkage to Cu⁷⁴ and electron withdrawing nature of the carboxylic acid end group would both operate to increase the surface potential contribution to the NP ϕ . Consequently, a reduction in MPA packing density with increasing NP diameter could give rise to the

observed decrease in NP ϕ with increasing diameter. To investigate this possibility the CPD of OA capped Cu NPs on Si was measured, taking special care to ensure the sample was analysed quickly after removal from the glovebox: Supplementary Figure 17 (Whilst OA capped Cu NPs are much less stable than MPA capped Cu NPs, the rate of oxidation is much slower than the time taken to collect the KP-AFM image (Supplementary Figure 9a)). In contrast to the case of MPA the polarity of the Cu-primary amine bond⁷⁵ together with the electron donating character of the alkyl chain⁷² operate to reduce the ϕ of the Cu NP from its bare value, and so a reduction in ligand density with increasing NP diameter would result in an increase in the ϕ of OA capped Cu NPs. It is evident from the data in Supplementary Figure 17 that the direction and magnitude of the change in ϕ is in fact very similar to that measured for MPA capped Cu NPs, and so it can be concluded that the variation in CPD with NP diameter results from a size effect, rather than a variation in ligand density.'

Point 9: Would the authors predict based on the current system to have ϕ on the same order of magnitude with bare particles as reported in there comparison with Zhou et al?

Response 9: We would predict that the change in work function with Cu NP diameter would be the same for any given ligand provided the ligand density did not change with NP size. However, the absolute magnitude of the work function would be strongly dependant on whether the ligand increased or decreased the surface potential contribution to the work function as discussed on page 13 and 14 of the revised manuscript.

Point 10: A minor point, size change resulting from thiols exchanging oleylamine on the NP surface is unlikely to be simply described as 0.4 nm change in radial thickness (packing density and attachment) is only qualitative. Uncertainty of some kind should be reported from AFM measurements, because it is similar to the reported uncertainty due to tip effects (Table S1). This reviewer highlights this point because it represents the issues with, in this reviewer's opinion, the unsubstantiated claims outline above.

Response 10: As detailed in response 5 we have now substantiated our claim that Cu nanoparticles are not etched during ligand exchange with thiols. We have also now included in Figure 8 and Supplementary Figures 16 and 17 the estimated error in the measurement of Cu NP diameter that takes into account the roughness of the supporting substrate and tip shape effects. Figures 8, S16 and S17 are already very dense with data points, so we have referred to this error in each of the figure captions rather than including error bars on each point.

Reviewer #2 (Remarks to the Author):

The paper reports a fascinating finding on the stability of copper nanoparticles when coated with organic thiols. Interestingly the authors observe not only good stability of the particles in thiol solution but also that short chain acid thiols protected the copper against oxidation more than long chain thiols. Having prepared the copper nanoparticles they have considered the variation in work function with particle size. Whilst I enjoyed the paper and felt it had general interest I am of the opinion that two points require clarification prior to acceptance:

Point 1: In looking at the dissolution of copper in thiol containing solutions the solvent should have been tested for trace copper to prove no dissolution had occurred.

Response 1: We thank Reviewer 2 for raising this point. We have now directly tested the solvent residue for the two thiol ligands on which AFM height analysis was performed (plus one other to confirm our conclusions) using both inductively coupled plasma mass spectroscopy and high

resolution photoelectron spectroscopy. The findings of these additional experiments support our original hypothesis. These additional experiments are discussed on page 10 & 11 of the revised manuscript and the new data is given in Supplementary Figure 7.

Point 2: Given the range of sizes of copper nanoparticles in the system and assuming uniform slow oxidation of the samples I would expect the spectra in figure 4 to change shape with time, an explanation as to why this does not occur should be provided.

Response 2: There is in fact a very significant change in the shape of the plasmonic resonance in Figure 4 a (that is consistent with Mie theory (Ref. 7, 18)) although we accept that this is difficult to see due to the overlapping spectra. Conversely the spectra in Figure 4 b changes very little which is consistent with one of the key findings of our paper: namely, that 3-mercaptopropionic acid is exception in its ability to slow oxidation of Cu NPs in air. We have now included in the Supplementary Information the absorption spectra at the beginning and after 16 hours air exposure only, so that any shape changes after 16 hours air oxidation can be seen more easily. We thank Reviewer 2 for raising this point.

Point 3: Two minor points:

1. Line 226 - missing unit on 11.7
2. Line 279 - remove at

Response 3: These errors have now been corrected.

Reviewers' comments:

Reviewer #1 (Remarks to the Author):

The authors have presented the data previously requested from this author, which in this author's opinion is necessary for full evaluation on the origin of stability. Some interpretation of the data is not aligned with what is known in the field about the passivation of NP surfaces. I have outlined below a few points for the authors to consider.

- XRD data has no internal calibration and assessment of small (4 nm) particles crystallinity is completed. I would suggest changing the text in the 2nd paragraph of the results. Also, 'free of oxides from XPS' is likely just below the LOD for the instrument, where all metal surfaces will be terminated with some hydroxylated structure.

-The analysis of the solution with ICP-MS suggests uniform etching of species with highly crystalline structure. This is highly unlikely based on the size distribution present. Etching could occur nonuniformly on some population of species (likely the smallest sizes of lower crystalline composition). Also, current ICP-MS instruments LOD detection for Cu is 3 orders of magnitude better than the reported numbers from the authors (1 ppt). This author does appreciate the honesty in presenting possible origins for the measured Cu in the solvents.

-The presented trend of packing density and ligand structure is weakly supported. The alkyl chain length argument only holds for two ligands. I would not consider that a trend. Also, surface passivation and oxygen diffusivity to the surface are handwaving at best. In the same paragraph on pg 14 that the authors say longer ligand length is better, the reversal trend is reported. A more cogent argument and presentation is necessary, or more succinct reporting of nontrends would be more appropriate.

-ref 70 is not appropriate to make a statement about ligand packing density as function of size. A large body of work has been devoted to this topic, which is in disagreement with the statement on page 21. I would refer the authors to work by the V. Rotello's group and others on this topic.

Reviewer #2 (Remarks to the Author):

The authors should be congratulated on performing the additional experiments requested by both referees. The additional experiments support the conclusions reached in the original manuscript and I now fully support the publication of this paper.

Reviewer comments and responses:

Reviewer 1:

The authors have presented the data previously requested from this author, which in this author's opinion is necessary for full evaluation on the origin of stability. Some interpretation of the data is not aligned with what is known in the field about the passivation of NP surfaces. I have outlined below a few points for the authors to consider.

Response: We thank Reviewer 1 for his/her comments on our revised manuscript, which we have now addressed in the revised version. Detailed responses to each point raised are given below together with the changes to the manuscript (highlighted in red).

Point 1: XRD data has no internal calibration and assessment of small (4 nm) particles crystallinity is completed. I would suggest changing the text in the 2nd paragraph of the results. Also, 'free of oxides from XPS' is likely just below the LOD for the instrument, where all metal surfaces will be terminated with some hydroxylated structure.

Response 1: It is evident from the AFM height analysis in Figure 3 that particles with a diameter ≤ 4 nm represent only a small proportion of the total sample (~5%). Whilst it is possible that this small population may have a different degree of crystallinity to the bulk of the population, the key findings of this work do not hinge on the degree of crystallinity in this very small population of nanoparticles. Given the above, and that Reviewer 1 did not raise this point in his/her first set of comments, we struggle to see how this point undermines the key findings reported in our manuscript.

As suggested we have now amended the second paragraph of the results section (specifically the first and last lines – reproduced below) to ensure the statements are more conservative and would like to thank Reviewer 1 for this suggestion:

'The X-ray diffraction (XRD) pattern of a thick film of Cu NPs shown in Fig. 2c confirms the high degree of crystallinity of the bulk of Cu NPs in the distribution: The reflections corresponding to Cu (111), Cu (200), Cu (220), Cu (311) and Cu (222) planes are assigned³⁴, although the possibility that the small population (~ 5%) of particles with

diameters less than ~ 4 nm may have lower crystallinity cannot be ruled out on the basis of these measurements.'

'XPS and Auger spectroscopy analysis confirm that freshly prepared OA capped and ligand exchanged CuNPs are essentially free of oxides (although the possibility of an extremely thin layer of oxide species just below the resolution of the instrument cannot be ruled out).'

Point 2: (i) The analysis of the solution with ICP-MS suggests uniform etching of species with highly crystalline structure. This is highly unlikely based on the size distribution present. Etching could occur nonuniformly on some population of species (likely the smallest sizes of lower crystalline composition). **(ii)** Also, current ICP-MS instruments LOD detection for Cu is 3 orders of magnitude better than the reported numbers from the authors (1 ppt). **(iii)** This author does appreciate the honesty in presenting possible origins for the measured Cu in the solvents.

Response 2: (i) We are surprised that Reviewer 1 should attach such significance to a simplified scenario given in the discussion to illustrate how little trace copper is actually found in the solvent solution after ligand exchange (also only in one case), and for which there could be number of possible explanations, the most likely of which has nothing to do with etching. This is particularly so, because in the same example we have also made the conservative assumption that only the top layer of copper nanoparticles in the multilayer film is etched, which further highlights how little copper (2.5 parts per billion) there is in the solvent. Furthermore, the nanoparticle height analysis presented in Figure 3, does not support Reviewer 1's hypothesis that the smallest sizes of lower crystalline composition may have been preferentially etched. Surely, what is important here is that the overwhelming body of evidence (based on the results of three different types of measurement; namely, ICPMS, XPS and AFM) shows that copper nanoparticles are not etched to any significant extent (and most probably not at all) when performing ligand exchange with thiols even for a ligand exchange time of 24 hours, which is well beyond the time actually required to achieve ligand exchange and so is a particularly harsh test.

(ii) ICPMS analysis performed using a higher resolution instrument than that available to us at this time **would add nothing to the interpretation of the data in this case**, because (as explained in the discussion) the possibility that some copper nanoparticles

are dislodged from the sample surface during the ligand exchange reaction can never be completely ruled out.

(iii) It is not a question of honesty, but of scientific rigor. The authors very much appreciate the time taken by Reviewer 1 to read and comment on our manuscript, and have gone to great lengths to address his/her comments by performing more than 12 new XPS/ICPMS experiments (to support non-etching of Cu nanoparticles and the generality of ligand exchange) and 1 new Kelvin probe AFM (to support the findings pertaining to the size effect of nanoparticle work function) the results of which are included in 21 pages of extra supporting information. Crucially we feel strongly that all of the new data support the original central findings of our manuscript – a view that is shared by Reviewer 2. We therefore request that the merits of our central findings are judged based on all of the evidence. It is always possible to find small points to criticise in any piece of work.

Point 3: The presented trend of packing density and ligand structure is weakly supported. The alkyl chain length argument only holds for two ligands. I would not consider that a trend. Also, surface passivation and oxygen diffusivity to the surface are handwaving at best. In the same paragraph on pg 14 that the authors say longer ligand length is better, the reversal trend is reported. A more cogent argument and presentation is necessary, or more succinct reporting of nontrends would be more appropriate.

-

Response 3: The discussion on page 14 was expanded at the request of Reviewer 1 in response to his/her Point 7 of the original set of comments which begins; 'Although not a requirement for the current work,..... (given in full below).' It is therefore puzzling to us as to why Reviewer 1 has now attached such importance to the simple observations (with reference to prior art) that we have included in the discussion. However, as requested we have now modified the text to give a more succinct reporting of non-trends, which in no way detracts from the central findings.

The discussion relating to the observations relating to alkyl thiols has been reduced to: '*Analysis of the Cu:S ratio determined from the XPS data (Supplementary Information Table 3) shows that for the simple alkylthiols the ligand packing density increases with increasing alkyl chain length, consistent with the expectation that a longer alkyl chain forms a more dense capping layer due to attractive inter-chain van der Waals*

interactions^{18,19,52}. *However, there is no trend in the stability towards air oxidation with increasing alkyl chain length since translating from 1-octanethiol (6) to 1-decanethiol (4) the resistance to oxidation increases, but then decreases for octadecanethiol (2).*

The last sentence in the first paragraph on 14 has also been restructured to read:
The importance of the end group functionality in determining the effectiveness of the ligand towards blocking oxidation of Cu NPs is highlighted by the fact that the stability of Cu NPs capped with MPA and 5 is similarly very high, although there is a significant difference in the thiol packing density between these two ligands.

Reviewer 1 (Point 7 from original set of comments):

Although not a requirement for the current work, the authors posit reasons for observed stability based on optical properties. However, again, this highlights the need to better characterize the surface of the NPs (e.g., coverage). On page 12-13, the authors provide observational differences based on chain length or terminus. However, these speculative claims could be directly investigated with spectroscopic methods, including XPS. These should be included in the work to augment the impact to the field for further advancing design principles.

Point 4: -ref 70 is not appropriate to make a statement about ligand packing density as function of size. A large body of work has been devoted to this topic, which is in disagreement with the statement on page 21. I would refer the authors to work by the V. Rotello's group and others on this topic.

Response: On page 21 of the revised manuscript, we have explicitly considered **both** reported possibilities as to how the ligand packing density varies as a function of nanoparticle size, with reference to a very recent review article on this topic; reference 70. Crucially, regardless of which is true for the case of copper nanoparticles (which to our knowledge is not yet actually known in the case of copper nanoparticles) the conclusions we have drawn as to how the work function of metallic nanoparticles scales with size, still hold.

As requested we have now replaced the offending reference with the review article; *Analyst*, 142, 11 (2017) (reference 71 in the revised manuscript) and added a new reference to support the more widely held view that ligand density decreases with increasing NP diameter (in addition to *Analyst*, 142, 11 (2017)).

We thank Reviewer 1 for his/her remarks.

Reviewer #2:

The authors should be congratulated on performing the additional experiments requested by both referees. The additional experiments support the conclusions reached in the original manuscript and I now fully support the publication of this paper.

Reviewer Comment: Reviewer #3 raised only one small technical point, which I paraphrase here: On p. 11 of the paper the optical dielectric constants should be used rather than the static dielectric constants in the discussion. This does not really affect the conclusion of the paper.

Response: As requested the optical dielectric constants of Cu₂O and CuO have now been given, with the addition of two references. The relevant sentence now reads (changes in red):

'The relatively large dielectric constants of Cu₂O and CuO (6-8)^{7,45,46}.....'

The references to the dielectric constant of ethanol and 1-octadecane thiol have been removed because they are superfluous to requirements: Regardless of their actual values it is obvious that the gradual evaporation of ethanol incorporated into the 1-octadecane ligand layer is on its own enough to account for the reduction in dielectric constant. In this way there is no increase in the total number of references and the sentence (which outlines only a tentative explanation) is now more succinct.

The original sentence read: '..... is tentatively attributed to the gradual evaporation of ethanol molecules trapped between the alkyl chains of the ligand, since ethanol has a much greater dielectric constant than 1-octadecanethiol (~ 1.8 vs. ~ 2.6⁴⁸) and is known to be easily trapped in monolayers of dodecanethiol^{49,50}.

The revised sentence reads: '.....**is tentatively attributed to the gradual evaporation of ethanol molecules trapped between the alkyl chains of the ligand, since ethanol is known to be easily trapped in monolayers of dodecanethiol^{48,49}.**'